# Structure-Activity Relationships of Holothuroid’s Triterpene Glycosides and Some In Silico Insights Obtained by Molecular Dynamics Study on the Mechanisms of Their Membranolytic Action

**DOI:** 10.3390/md19110604

**Published:** 2021-10-25

**Authors:** Elena A. Zelepuga, Alexandra S. Silchenko, Sergey A. Avilov, Vladimir I. Kalinin

**Affiliations:** G.B. Elyakov Pacific Institute of Bioorganic Chemistry, Far Eastern Branch of the Russian Academy of Sciences, Pr. 100-letya Vladivostoka 159, Vladivostok 690022, Russia; zel@piboc.dvo.ru (E.A.Z.); silchenko_als@piboc.dvo.ru (A.S.S.); avilov_sa@piboc.dvo.ru (S.A.A.)

**Keywords:** triterpene glycosides, sea cucumber, membranolytic action, hemolytic, cytotoxic activity, molecular dynamic simulation

## Abstract

The article describes the structure-activity relationships (SAR) for a broad series of sea cucumber glycosides on different tumor cell lines and erythrocytes, and an in silico modulation of the interaction of selected glycosides from the sea cucumber *Eupentacta fraudatrix* with model erythrocyte membranes using full-atom molecular dynamics (MD) simulations. The in silico approach revealed that the glycosides bound to the membrane surface mainly through hydrophobic interactions and hydrogen bonds. The mode of such interactions depends on the aglycone structure, including the side chain structural peculiarities, and varies to a great extent. Two different mechanisms of glycoside/membrane interactions were discovered. The first one was realized through the pore formation (by cucumariosides A_1_ (**40**) and A_8_ (**44**)), preceded by bonding of the glycosides with membrane sphingomyelin, phospholipids, and cholesterol. Noncovalent intermolecular interactions inside multimolecular membrane complexes and their stoichiometry differed for **40** and **44**. The second mechanism was realized by cucumarioside A_2_ (**59**) through the formation of phospholipid and cholesterol clusters in the outer and inner membrane leaflets, correspondingly. Noticeably, the glycoside/phospholipid interactions were more favorable compared to the glycoside/cholesterol interactions, but the glycoside possessed an agglomerating action towards the cholesterol molecules from the inner membrane leaflet. In silico simulations of the interactions of cucumarioside A_7_ (**45**) with model membrane demonstrated only slight interactions with phospholipid polar heads and the absence of glycoside/cholesterol interactions. This fact correlated well with very low experimental hemolytic activity of this substance. The observed peculiarities of membranotropic action are in good agreement with the corresponding experimental data on hemolytic activity of the investigated compounds in vitro.

## 1. Introduction

The majority of triterpene glycosides from sea cucumbers possess strong hemolytic and cytotoxic actions against different cells, including cancer cells [1,2,3,4]. However, the mechanism of their membranolytic action is not yet fully understood at the molecular level, particularly in relation to the structural diversity of these compounds. Some trends of SAR of sea cucumber glycosides have been discussed [5,6], but the molecular interactions of different functional groups with the components of biomembranes which affect the membranotropic action of the glycosides remain unexplored.

The broad spectrum of bioactivity of sea cucumber triterpene glycosides derives from their ability to interact with the lipid constituents of the membrane bilayer, changing the functional properties of the plasmatic membrane. Sterols are very important structural components influencing the properties and functions of eukaryotic cell membranes. The selective bonding to the sterols of the cell membranes underlines the molecular mechanisms of action of many natural toxins, including triterpene glycosides of the sea cucumbers. The formation of complexes with 5,6-unsaturated sterols of target cell membranes is the basis of their biological activity including ichthyotoxic action that may protect sea cucumbers against fish predation. In fact, some experimental data indicated the interaction of the aglycone part of the glycosides with cholesterol [7,8]. The saturation of ascites cell membranes with cholesterol increased the cytotoxicity of the sea cucumber glycosides [9]. This complexing reaction of both the animal and plant saponins leads to the formation of pores, the permeabilization of cells, and in the case of red blood cells for which the membranes are known to be enriched in cholesterol [10], the subsequent loss of hemoglobin in the extracellular medium [11]. Malyarenko et al. tested a series of triterpene glycosides isolated from the starfish *Solaster pacificus* that had exogenic origin from a sea cucumber eaten by this starfish [12]. The authors showed that the addition of cholesterol to corresponding tumor cell culture media significantly decreases the cytotoxicity of these glycosides. It clearly confirmed the cholesterol-dependent character of the membranolytic action of sea cucumber triterpene glycosides. It is of special interest that the activity of a glycoside with 18(16)-lactone instead of 18(20)-lactone, and a shortened side chain, was also decreased by the adding of cholesterol.

The sea cucumber glycosides may be active in subtoxic concentrations, and such a kind of activity is cholesterol-independent. Aminin et al. showed that the immunostimulatory action of cucumarioside A_2_-2 from *Cucumaria japonica* resulted from the specific interaction of the glycoside with a P2X receptor and was cholesterol-independent [13]. The addition of cholesterol to the medium or to the mixture of substances may decrease the cytotoxic properties of the glycosides while preserving their other activities. This property of cholesterol has been applied to the development of ISCOMs (immune-stimulating complexes) and subunit protein antigen-carriers, composed of cholesterol, phospholipid, and glycosides [14,15]. Moreover, the immunomodulatory lead–“Cumaside” as a complex of monosulfated glycosides of the Far Eastern Sea cucumber *Cucumaria japonica* with cholesterol, has been created [16]. It possesses significantly less cytotoxic activity against sea urchin embryos and Ehrlich carcinoma cells than the corresponding glycosides, but has an antitumor activity against different forms of experimental mouse Ehrlich carcinoma in vivo [17].

Therefore, cholesterol seems to be the main molecular target for the majority of glycosides in the cell membranes. However, the experimental data for some plant saponins indicate that saponin-membrane binding can occur independently of the presence of cholesterol, cholesterol can even delay the cytotoxicity, such as for ginsenoside Rh2, and phospholipids or sphingomyelin play an important role in these interactions [7,18]. Thus, different mechanisms exist, cholesterol-dependent and -independent, that are involved in saponin-induced membrane permeabilization, depending on the structure of saponins [11]. However, recent in vitro experiments and the monolayer simulations of membrane binding of the sea cucumber glycoside frondoside A, confirmed previous findings that suggest the presence of cholesterol is essential to the strong membranolytic activity of saponins. However, the cholesterol-independent, weak binding of the glycoside to the membrane phospholipids, driven by the lipophilic character of the aglycone, was discovered. Then saponins assemble into complexes with membrane cholesterol followed by the accumulation of saponin-sterol complexes into clusters that finally induce curvature stress, resulting in membrane permeabilization and pore formation [7].

The aims of this study were: the analysis of SAR data for a broad series of sea cucumber glycosides, mainly obtained by our research team over recent years on different tumor cell lines and erythrocytes and additionally the explanation for these data by modelling the interactions of the glycosides from the sea cucumber *Eupentacta fraudatrix* with the constituents of model red blood cell membrane through the full-atom molecular dynamics (MD) simulation. Such an investigation appears even more relevant since different molecules and their complexes composing of cell membranes, for example, cholesterol-enriched lipid rafts, have continued to attract attention as the factors involved in tumorigenesis and a number of cellular pathways related to cell survival, proliferation, and apoptosis [19]. Therefore, it may be of great advantage to modulate the interactions of the membrane constituents by membrane-active compounds, such as triterpene glycosides. An in silico technique was applied to reinforce the numerous experimental observations (SAR) by modelling a multitude of inter-molecular interactions at a high spatial (atomic level) and temporal (nanosecond) resolution within a simulation framework that can reconstitute the natural behavior on the basis of physical interactions [20,21]. Such in silico MD simulations can be regarded as a “computational microscope” capable of visualizing molecular behavior with unprecedented precision [22].

## 2. Results and Discussion

### 2.1. Structure-Activity Relationships (SAR) Observed in the Glycosides from Sea Cucumbers

The triterpene glycosides of sea cucumbers are natural compounds that have been investigated for a long time. Several hundred structures of the glycosides are now known from the representatives of different orders of the class Holothuroidea. The finding of a significant number of new glycosides by our research team, especially over recent years, led to the broadening of the knowledge of their great structural diversity. This facilitated SAR highlighting through the comparative analysis of their structural features, including both carbohydrate chain composition and architecture, aglycone structures, and their bioactivity (cytotoxic and hemolytic action).

It became obvious that glycosides cytotoxic activity depends not only on their structures, but also on the type of processed cells differing by composition and functional peculiarities of their membranes [1]. Analyzing the majority of tested compounds, it has been revealed that the membranes of erythrocytes are more sensitive to the glycoside membranolytic action than the other tested cells such as mouse spleen lymphocytes, ascites of mouse Ehrlich carcinoma, and neuroblastoma Neuro 2a or normal epithelial JB-6 cells. This regularity was observed in the glycosides of the *Actinocucumis typica* [23], *Colochirus robustus* [24], *Massinium magnum* [25,26], *Eupentacta fraudatrix* [27,28,29], *Psolus fabricii* [30,31], and *Colochirus quadrangularis* [32] sea cucumbers. One of the explanations for this phenomenon may be the enrichment of red blood cell membranes with cholesterol [10].

The structures of holothuroids’ triterpene glycosides vary in a number of structural features while retaining the general plan of molecular structure. The influence of structural signs such as the monosaccharide composition and architecture of carbohydrate chains, the quantity and positions of sulfate groups, the type of aglycone, and the structure of a side chain on the activity of the glycosides is significant.

#### 2.1.1. The Dependence of the Glycosides Hemolytic Activity on Their Carbohydrate Chain Structure

It was earlier noticed that the presence of a linear tetrasaccharide chain is necessary for the membranolytic action of the glycosides, that glycosides with quinovose as the second sugar unit in the chain are more active than those with glucose or xylose, and that the sulfate group at C-4 Xyl1 increases the activity of tetraosides and pentaosides with sugar parts branched by C-2 Qui2, however the sulfate groups at C-6 Glc3 and C-6 MeGlc4 of such pentaosides significantly decrease the activity [4,33,34].

In fact, the comparison of the hemolytic activities (Table 1) of cucumariosides B_1_ (**1**) and B_2_ (**2**) from *E. fraudatrix* with a trisaccharide chain with the monosaccharide residues attached to each other by β-(1➝2)-glycosidic bonds [35], and the activity of cucumariosides H_5_ (**3**) and H (**4**) (Figure 1) with pentasaccharide monosulfated chains [36] revealed the significance of the linear tetrasaccharide fragment with terminal *O*-methylated sugar residue, as the compounds **1**, **2** were almost not active.

The analysis of the data on the hemolytic activity of the glycosides from *M. magnum,* with identical aglycones and differing by the oligosaccharide chain structures, demonstrates that the influence of the carbohydrate chain structure indirectly depends on its combination with different aglycones. There were three groups of the glycosides in *M. magnum*: monosulfated biosides (magnumosides of the group A (**5**–**7**)), monosulfated tetraosides (magnumosides of the group B (**8**–**11**)) and disulfated tetraosides (magnumosides of the group C (**12**–**15**)) (Figure 2), all were attached to non-holostane aglycones with 18(16)-lactone differing by the side chain structures [25,26].

In the series of magnumosides B_1_ (**8**) and C_1_ (**12**) and magnumosides A_2_ (**5**), B_2_ (**9**), C_2_ (**13**), with the hydroxyl group in the aglycone side chains, the disulfated tetraosides **12** and **13** were the most active compounds, while in the series of magnumosides A_3_ (**6**), B_3_ (**10**), C_3_ (**14**) and magnumosides A_4_ (**7**), B_4_ (**11**), C_4_ (**15**), which comprised the side chains with a double bond, the monosulfated tetraosides **10** and **11** showed the strongest effect (Table 1). Magnumosides of group A (**5**–**7**) demonstrated considerable hemolytic effects despite the absence of a tetrasaccharide linear fragment (Table 1). A compensation for the absence of two sugars by a sulfate at C-4 of the first xylose residue was earlier described for sea cucumber glycosides with 18(20)-lactone in aglycones. [5,33].

The interesting observations were made when the activity of the glycosides from the sea cucumber *Psolus fabricii* (Figure 3) was analyzed [30,31]. Psolusosides A (**16**) and E (**17**) having linear tetrasaccharide sugar moieties were the strongest cytotoxins in this series, but the activity of psolusosides H (**18**) and H_1_ (**19**) (the glycosides with trisaccharide chains) was close to that of the linear tetraosides **16**, **17** (Table 1) despite the absence of tetrasaccharide linear moiety and the change in the second unit (quinovose) in the chain of **16**, **17** to glucose residue in **18**, **19**. However, psolusosides J (**20**) and K (**21**) with tetrasaccharide chains branched by C-4 Xyl1 and three sulfate groups were completely inactive despite the presence of holostane (i.e., with 18(20)-lactone) aglycones.

The majority of the glycosides found in the sea cucumber, *Cladolabes schmeltzii,* and characterized by penta- or hexasaccharide moieties branched by C-4 Xyl1, demonstrated strong hemolytic action that was only slightly dependent on their monosaccharide composition. The general trend observed was that hexaosides are more active than pentaosides [37,38,39,40,41].

Therefore, the influence of carbohydrate chain structure on the activity of glycosides is mediated by its combination with the aglycone, however, the general trend is that more developed (tetra-, penta- and hexa-saccharide) sugar moieties provide higher membranolytic action.

#### 2.1.2. The Dependence of Hemolytic Activity of the Gycosides on the Positions and Quantity of Sulfate Groups

The comparison of the hemolytic effects of typicosides B_1_ (**22**) and C_2_ (**23**) from *A. typica* [23] (Figure 4)–linear tetraosides differing by the quantity of sulfate groups showed that the disulfated compound **23** is more active than a monosulfated one (Table 1). High hemolytic activity was demonstrated by the sulfated glycosides from *C. shcmeltzii* [39]–cladolosides of groups I (**24**, **25**) and J_1_ (**26**), with pentasaccharide chains branched by C-4 Xyl1 with the sulfate group at C-6 MeGlc in the bottom or upper semi-chains, correspondingly, as well as cladolosides K_1_ (**27**) and L_1_ (**28**)–with monosulfated hexasaccharide chains differing by the sulfate group position (Figure 4). This trend was also confirmed by SAR demonstrated by the glycosides from *P. fabricii* [31]. Psolusoside L (**29**) (Figure 5) was strongly hemolytic in spite of the presence of three sulfate groups at C-6 of two glucose and 3-O-methylglucose residues in the pentasaccharide chain branched by C-4 Xyl1. Thus, the presence of sulfate groups attached to C-6 of monosaccharide units did not decrease the activity of pentaosides branched by C-4 Xyl1 in comparison to that of pentaosides branched by C-2 Qui2 [4,33].

The influence of sulfate position is clearly reflected through the comparison of the activity of psolusosides M (**30**) and Q (**31**). The latter glycoside was characterized by the sulfate position attached to C-2 Glc5 (the terminal residue), that caused an extreme decrease in its activity (Table 1). Even the tetrasulfated (by C-6 Glc3, C-6 MeGlc4, C-6 Glc5, and C-4 Glc5) psolusoside P (**32**) was much more active than trisulfated psolusoside M (**30**) containing the sulfate group at C-2 Glc5 (Figure 5).

The analysis of SAR in the raw of glycosides from the sea cucumbers *Colochirus quadrangularis* [32] (quadrangularisosides B_2_ (**33**), D_2_ (**34**), and E (**35**)), *C. robustus* [24] (colochiroside C (**36**)) (Figure 6) and *P. fabricii* [30] (psolusosides A (**16**), E (**17**) (Figure 3), and F (**37**)) (Figure 6) with the same holostane aglycone and linear tetrasaccharide chains and differing by the third monosaccharide residue and the number and positions of sulfate groups, showed that they all were strong hemolytics (Table 1). However, the presence of a sulfate group at C-4 or C-6 of terminal MeGlc residue resulted in approximately a tenfold decrease in activity, while the sulfation of C-3 Qui2 or C-6 Glc3 did not decrease the hemolytic action.

Hence, the influence of sulfate groups on the membranolytic action of triterpene glycosides depends on the architecture of their carbohydrate chains and the positions of attachment of these functional groups.

#### 2.1.3. The Dependence of Hemolytic Activity of the Glycosides on Aglycone Structure

In the earlier studies of glycoside SAR, the necessity of the presence of a holostane-type aglycone (with 18(20)-lactone), was noticed for the compound to be active. The glycosides containing non-holostane aglycones (i.e., having 18(16)-lactone, without a lactone with a shortened or normal side chain), as a rule, demonstrate only weak membranolytic action [4,33]. However, different functional groups attached to polycyclic nucleus or the side chain of holostane aglycones can significantly influence the membranotropic activity of the glycosides.

All the glycosides isolated from *M. magnum* contain non-holostane aglycones with 18(16)-lactone, 7(8)-double bond and a normal (non-shortened) side chain. Despite this fact, the compounds demonstrated high or moderate hemolytic effects (Table 1) (except for the compounds containing OH-groups in the side chains) [25,26]. Nevertheless, the comparison of hemolytic activity of the pairs of compounds (Table 1) colochiroside B_2_ (**38**) (Figure 7) and magnumoside B_1_ (**8**), as well as colochiroside C (**36**) and magnumoside C_3_ (**14**), and differing by the aglycones nuclei (holostane and non-holostane, correspondingly), showed that compounds **36** and **38**, which contained the holostane aglycones, were more active, and this is consistent with the earlier conclusions.

Additionally, the glycosides of the sea cucumber, *Cucumaria fallax* [42], did not display any activity due to containing unusual hexa-*nor*-lanostane aglycones with an 8(9)-double bond and without a lactone. The only glycoside from this series, cucumarioside A_3_-2 (**39**) (Figure 8), that was moderately hemolytic (Table 1) was characterized by hexa-*nor*-lanostane aglycone, but, as typical for the glycosides of sea cucumbers, having a 7(8)-double bond and 9*β*-H configuration, which demonstrates the significance of these structural elements for the membranotropic action of the glycosides.

The influence of the side chain length and character of a lactone (18(20)- or 18(16)-) is nicely illustrated by the comparative analysis of the hemolytic activity of the series of glycosides from *E. fraudatrix* (cucumariosides A_1_ (**40**) and A_10_ (**41**) [28,29]; cucumariosides I_1_ (**42**) and I_4_ (**43**) [43]) (Figure 9), which indicates that the presence of a normal side chain is essential for the high membranolytic effect of the glycoside.

Unexpectedly high hemolytic activity was displayed by cucumarioside A_8_ (**44**) from *E. fraudatrix* [29] (Figure 10) with unique non-holostane aglycone and without lactone but with hydroxy-groups at C-18 and C-20, which can be considered as a biosynthetic precursor of the holostane aglycones. Its strong membranolytic action (Table 1) could be explained by the formation of an intramolecular hydrogen bond between the atoms of aglycone hydroxyls resulting in the spatial structure of the aglycone becoming similar to that of holostane-type aglycones. Noticeably, it is of special interest to check this issue by in silico calculations to clarify the molecular mechanism of membranotropic action of **44**.

#### 2.1.4. The Influence of Hydroxyl Groups in the Aglycones Side Chain to Hemolytic Activity of the Glycosides

A strong activity-decreasing effect of the hydroxyl groups in the aglycone side chains was revealed for the first time when the bioactivity of the glycosides from *E. fraudatrix* was studied [27,28,29,43]. In fact, cucumariosides A_7_ (**45**), A_9_ (**46**), A_11_ (**47**), and A_14_ (**48**), as well as I_3_ (**49**), were not active against erythrocytes (Table 1) (Figure 11).

However, colochirosides B_1_ (**50**) (Figure 11) and B_2_ (**38**) from *C. robustus* [24], with the same aglycones as cucumariosides A_7_ (**45**) and A_11_ (**47**), correspondingly, but differing by the third (Xylose) and terminal monosaccharide residues (3-O-MeGlc) and the presence of sulfate group at C-4 Xyl1, demonstrated moderate hemolytic activity (Table 1). The activity of typicoside C_1_ (**51**) from *A. typica* [23] as well as cladolosides D_2_ (**52**) and K_2_ (**53**) from *C. schmeltzii* [40,41], with a 22-OH group in the holostane aglycones, was significantly lower (Table 1) than that of typicoside C_2_ (**23**) and cladolosides D_1_ (**54**) (Figure 12) and K_1_ (**27**), correspondingly, containing a 22-OAc group.

The same activity-decreasing effect of the hydroxy-group in the side chain was observed for the glycosides of *M. magnum* with non-holostane aglycones (monosulfated glycosides (**5**, **8**, **9**)), however, the presence of additional sulfate groups in the carbohydrate chains of **12**, **13** compensated this influence to some extent (Table 1).

Recently, the glycosides containing hydroperoxyl groups in the aglycone side chains were found in sea cucumbers *C. quadrangularis* (quadrangularisosides A (**55**) and A_1_ (**56**) [32]) and *P*. *fabricii* (psolusoside D_3_ (**57**) [44]) (Figure 13). The comparative analysis of their hemolytic activity with that of their structural analogs that contained hydroxyl groups in the same positions (colochirosides B_2_ (**38**), B_1_ (**50**), and psolusoside D_5_ (**58**), correspondingly) showed that the influence of OOH-functionalities was not so negative (Table 1).

#### 2.1.5. Correlation Analysis

To determine the structural elements of glycosides that might be responsible for membrane recognition, a set of physical properties of fifty-nine glycosides (represented in Table 1) were analyzed with MOE 2020.0901 CCG software [45]. Models of the spatial structure of the studied glycosides were built, protonated at pH 7.4, and subjected to energy minimization and a conformational search with MOE 2020.0901 CCG, and the dominant glycoside conformations were selected. The numerical descriptions or characterizations of the molecules that provide their physical properties such as the octanol/water partition coefficient, the polar surface area, the van der Waals (VDW) volume, the approximation to the sum of VDW surface areas of pure hydrogen bond acceptors/donors, the approximation to the sum of VDW surface areas of hydrophobic/polar atoms, etc. (296 in total), as well as their correlation matrix, were calculated with the QuaSAR-Descriptor tool of MOE 2020.0901 CCG software [45] (Appendix A). The correlation analysis did not reveal any strong direct correlation between the hemolytic activities of these compounds in vitro (Table 1) and certain calculated molecular 2D and 3D descriptors. Nevertheless, moderate positive correlations of their activity with the atomic contribution of the octanol/water partition coefficient [46], the total negative VDW surface area (Å^2^), the number of oxygen atoms, the atomic valence connectivity index, the kappa shape indexes [47], which describe the different aspects of molecular shape, and the molecular VDW volume (Å^3^) were disclosed. Therefore, an obvious joint effect of the molecular shape and volume (including the carbohydrate moiety shape and volume), the negative charge surface distribution, and the oxygen atom content on the membranotropic properties of the glycosides was observed. These results indicate the extremely complex nature of relationships between the structure of glycosides and their membranolytic action.

The analysis of SAR of the broad series of the glycosides from sea cucumbers also confirms the complicated and ambiguous character of these relationships because the impact in the membranotropic action of a certain structural element depends on the combination of such elements in the glycoside molecule. Nevertheless, there are some structural features causing the activity of the glycosides to be significant:The presence of a developed carbohydrate chain composed of four to six monosaccharide residues or a disaccharide chain with a sulfate group;The availability of 18(20)- or 18(16)-lactone and a normal (non-shortened) side chain;The presence of 9*β*-H, 7(8)-ene fragment, or 9(11)-double bond.

The influence of sulfate groups on the membranotropic action of the glycosides depends on the architecture of the sugar chain and the positions of sulfate groups. Hydroxyl groups attached to different positions of aglycone side chains extremely decrease the activity.

### 2.2. In Silico Analysis of the Interaction of the Glycosides from the Sea Cucumber Eupentacta fraudatrix with the Model Membrane

The molecular mechanisms of action of membranotropic compounds to the natural cell membranes are difficult to observe directly with any experimental techniques. Moreover, the lipid composition of membranes of diverse eukaryotic cell types varies to a great extent. The MD simulation providing information at the molecular level has become an increasingly popular “molecular-specific” technique for the study of issues related to bioactive molecule interactions with the membranes due to the rise of computing power, the development of methodologies, software, as well as the force field parameters. Nevertheless, artificial lipid bilayer membranes are suitable models for such investigations providing results consistent with the data obtained in the experiments with different cell lines. In this investigation the lipid composition of model membrane was chosen taking into account the balance between its complexity (resemblance to reality) and the feasibility of biophysical observations to interpret. The model of the symmetrical bilayer membrane containing two or three lipid types (phosphatidylcholine, sphingolipid, and sterol) is the most frequently used. Therefore, the artificial model of the erythrocyte-mimicking membrane constituting of phosphatidylcholine (POPC), cholesterol (CHOL), and palmitoyl-sphingomyelin (PSM) or 1-palmitoyl-2-oleoyl-sn-glycero-3-phosphoethanolamine (POPE) for the outer or inner leaflet, respectively, in a saline solution environment, was constructed based on the lipid composition of red blood cell membranes known to contain approximately 48% CHOL, 28% phosphatidylcholine and 24% sphingomyelin in the outer membrane leaflet, as well as phosphoethanolamine in the inner membrane leaflet [10,48].

To determine the membrane molecular targets of the binding glycosides, the simulations of full-atom molecular dynamic (MD) for the interactions of cucumariosides A_1_ (**40**), A_2_ (**59**), A_8_ (**44**), and A_7_ (**45**) from the sea cucumber *Eupentacta fraudatrix* (Figure 14) (hemolytic activities demonstrated by these compounds in vitro are presented in Table 1), differing from each other by the side chain or aglycone (for **44**) structures, with the model membrane for 600 ns time length (for each) were conducted (see Materials and Methods for details). The same MD simulations protocol was applied for the solvated bilayer system without the glycoside exposure, to be used as a control.

#### 2.2.1. The Modelling of Cucumarioside A_1_ (**40**) Membranotropic Action with MD Simulations

Our results derived from MD simulations of a model membrane system in the presence of cucumarioside A_1_ (**40**) demonstrated that glycoside is able to interact specifically with the PSM of the outer membrane leaflet. The analysis of intermolecular interactions (Figure 15A) of cucumarioside A_1_ (**40**), characterized by 24(25)-double bond, showed the attachment of its carbohydrate chain to membrane sphingomyelin (PSM) by hydrogen bonds (with the energy contribution of −11.94 kcal/M) (Table 2) enabling the anchoring of the glycoside at the interface of the membrane which is similar to dioscin behavior [49].

Further MD simulations in the system CHOL/POPC/PSM/POPE which was exposed to cucumarioside A_1_ (**40**) molecules demonstrated that glycoside integrates into the outer membrane leaflet leading to an asymmetrical pore formation with 7.52 Å diameter in the central part and 14.56 Å diameter in the entrance (Figure 15B,C). The stoichiometry of the pore forming components, glycoside/CHOL/POPC/PSM, is 2/4/5/6. Hence, cucumarioside A_1_ (**40**) is capable of incorporating into the outer membrane leaflet predominantly through hydrophobic interactions of its aglycone with phospholipids, sphingomyelin, as well as cholesterol, that results in the membrane curvature, followed by its destabilization and permeability changing. It should be noted that during the formation of this multimolecular pore-like structure induced by cucumarioside A_1_ (**40**), sphingomyelin molecules interact tightly with both glycosides and cholesterol through hydrogen-bonding as well as through hydrophobic interactions. Thus, sphingomyelin and cholesterol act as a functional pair to stabilize these complexes, similar to how they stabilize lipid rafts [22,50]. These data are in accordance with the high hemolytic effect of cucumarioside A_1_ (**40**) (Table 1).

The RMSD value of the heavy atoms of the model membrane phospholipids under cucumarioside A_1_ (**40**) action was 2.89 Å, while for the lipid environment surrounding the glycoside at 10 Å (POPC, CHOL, PSM) it was 4.13 Å. Moreover, the deviation of CHOL heavy atoms in the outer leaflet did not exceed 1.89 Å, while in the inner leaflet the RMSD value was 4.97 Å and reached up to 6.99 Å for some of CHOL molecules (Figure 15C). 

#### 2.2.2. The Modelling of Cucumarioside A_8_ (**44**) Membranotropic Action with MD Simulations

The in silico study of the action of cucumarioside A_8_ (**44**) from *E. fraudatrix* [29] on a model erythrocyte membrane with MD simulations evidenced that the process apparently occurs in several stages: driven by electrostatic attracting, the glycoside reaches the membrane with its carbohydrate part and can anchor to phospholipid polar heads through hydrogen bonds (Figure 16C and Appendix A), after that its aglycone moiety completely immerses into the lipid layer, and the multimolecular assembly rearranges. Moreover, our computational results have disclosed the feasibility of the glycoside to induce the “pore-like” complex formation inside the membrane with stoichiometry of glycoside/CHOL/POPC/PSM (2/3/2/5) (Figure 16A,B, Table 3). Its assemblage is provided mainly through van der Waals bonds and hydrophobic interactions with PSM and contributes totally to complex formation up to −62.07 kcal/M. Simultaneously, the aglycone of one glycoside molecule (I) is anchored to a PSM head by a hydrogen bond (contributing −1 kcal/M), whereas the carbohydrate moiety of the other molecule (II) stabilizes this complex by another hydrogen bond generated with a POPC molecule (with a contribution of −3.10 kcal/M) (Table 3). This suggests that the mechanism of cucumarioside A_8_ (**44**) hemolytic action is somewhat similar to that of cucumarioside A_1_ (**40**).

The important functional role of hydroxy groups at C-18 and C-20 of cucumarioside A_8_ (**44**) were found to promote an initial stage of glycoside integration into the lipid bilayer by the multiple hydrogen bond formations with sphingomyelin or phosphatidylcholine (Figure 16C). However, the extensive hydrophobic interactions became more energetically favorable at the subsequent stages of the glycoside engagement inside the outer membrane leaflet, allowing it to penetrate rather deeply into the bilayer (Appendix A). Moreover, further MD simulations have revealed the inner membrane leaflet rearrangement under the influence of cucumarioside A_8_ (**44**). Thus, the aglycone passed through the outer membrane leaflet and initiated the phosphatidylcholine molecule tails to move from the inner layer towards the “pore-like” assembly to generate hydrophobic interactions with the glycoside side chains (with a contribution of −3.72 kcal/M and −2.02 kcal/M) (Table 3, Appendix A).

The analysis of noncovalent intermolecular interactions in this complex shows that, in contrast to the pore formed by cucumarioside A_1_ (**40**), where the glycoside interacts predominantly with the lipid environment (CHOL/POPC/PSM) of the outer membrane layer (Table 2), the aglycone moieties of cucumarioside A_8_ (**44**) molecules formed rather powerful hydrophobic contacts between each other (with a contribution of −8.75 kcal/M), as well as hydrogen bonds between their carbohydrate parts, contributing approximately −3.49 kcal/M to the complex formation. Apparently, these glycoside/glycoside interactions inside the pore led to a decrease in its diameter to 13.06 Å in the entrance and 3.96 Å in its narrowest part as compared to those for the cucumarioside A_1_ (**40**)-induced pore (Figure 15). This finding suggests that the glycoside **44** is capable of forming pores in the erythrocyte membrane, similar to the glycoside **40**, but their size and quantity would be more sensitive to the glycoside concentration. This result is in good agreement with the glycoside activities (Table 1), indicating an order of magnitude higher hemolytic activity of cucumarioside A_1_ (**40**) compared to that of cucumarioside A_8_ (**44**).

#### 2.2.3. The Modelling of Cucumarioside A_2_ (**59**) Membranotropic Action with MD Simulations

MD simulations of interactions of cucumarioside A_2_ (**59**), with a 24-*O*-acetic group, demonstrated that glycoside bound to both the phospholipids and cholesterol of the outer membrane leaflet causing significant changes in the bilayer architecture and dynamics. The apolar aglycone part of the glycoside and the fatty acid residues of phospholipids interact with each other through hydrophobic bonds (with energy contribution from −1.23 kcal/M to −4.65 kcal/M) and hydrogen bonds (with energy contribution from −0.50 kcal/M to −8.20 kcal/M) (Table 4, Figure 17). The analysis of the energy contributions of different membrane components to the formation of multimolecular complexes including three molecules of cucumarioside A_2_ (**59**) revealed that the glycoside/phospholipid interactions were more favorable compared to the glycoside/cholesterol interactions involving only the aglycone side chain area (Figure 17). One molecule of the glycoside interacted with 3–5 phospholipid molecules involving their polar heads being bound to the polycyclic nucleus and carbohydrate chains while fatty acid tales surrounded the aglycones side chain. Thus, a so-called “phospholipid cluster” is formed around the glycoside causing it to be partly embedded to the outer leaflet. A rather rigid “cholesterol cluster” is formed under the place of glycoside penetration to the outer membrane leaflet due to the lifting of cholesterol molecules from the inner leaflet attempting, to some extent to substitute the molecules of the outer leaflet which are bound with the glycoside (Figure 17).

Therefore, the agglomerating action of cucumarioside A_2_ (**59**) towards the cholesterol molecules not only in the immediate vicinity of the glycoside but involving the cholesterol molecules from the inner membrane leaflet became clear. However, since cholesterol, with its rather rigid structure, interacts mainly with the aglycone side chain, it continues to be embedded to the outer leaflet, while flexible phospholipid molecules, interacting with both the aglycone and carbohydrate chain, to some extent overlook the outer membrane leaflet. Hence, two so-called “lipid pools” are generated with one of them surrounding carbohydrate and polycyclic moieties of the glycoside and the second one located in the aglycone side chain area (Figure 17B). 

Due to the asymmetric distribution of lipids between the membrane monolayers, their properties can differ significantly. POPC and PSM are characterized by saturated fatty acid tails, the asymmetry of leaflets is enhanced by different polar head properties of POPC, PSM, and POPE. Moreover, the presence of CHOL molecules in the bilayer, the content of which is close to 50% in the erythrocyte biomembrane, promotes the “elongation” and alignment of fatty tails of phospholipids parallel to the flat core of CHOL [51]. Our MD simulation results suggest that cucumarioside A_2_ (**59**) apparently induced the disruption of tight CHOL/lipid and lipid/lipid interactions through an extensive hydrophobic area formation in the glycoside’s immediate environment (Figure 17, Table 4). Additionally, the glycoside can provoke the process of CHOL release from the inner monolayer and its accumulation between monolayers or insertion to the outer one, because, unlike POPC, PSM and POPE, which have rather bulk polar heads, the small polar OH-group of CHOL is known to facilitate CHOL relocation between monolayers due to the low energy barrier of the “flip-flop” mechanism [51]. All these properties and forces led to the accumulation of CHOL molecules surround the glycoside, which resulted in an increase in layer viscosity. Simultaneously the CHOL outflow made the inner leaflet more fluid and unbalanced compared to the structured outer one that can cause the generation of non-bilayer disordered membrane architecture. These circumstances cause the inner membrane leaflet to be reorganized followed by the changing of the membrane barrier properties providing the hemolytic action of the glycoside.

Thus, according to our MD simulations, cucumarioside A_2_ (**59**) exposure caused significant change in the architecture of the model membrane bilayer (Figure 17). It should be noted that although the dynamic behavior of the lipid environment of cucumarioside A_1_ (**40**) and cucumarioside A_2_ (**59**) was similar, there were a number of considerable differences. Despite the low RMSD value for all heavy atoms of membrane lipids (3.74 Å), which reflects the mobility of membrane components, the dynamic behavior of those located in the immediate environment of the glycoside molecules was changed to a great extent. So, their RMSD value (10 Å surrounding the glycoside) was 7.47 Å, and for some CHOL molecules, predominantly those forming the inner membrane leaflet, this value reached 17.68 Å.

#### 2.2.4. The Modelling of Cucumarioside A_7_ (**45**) Membranotropic Action with MD Simulations

Cucumarioside A_7_ (**45**) differs from the compounds **40**, **44**, and **59** by the presence of an OH-group in the aglycone side chain that causes the extremal decreasing in its membranotropic activity (Table 1). In fact, in silico simulations of its interactions with model membrane demonstrated only slight interactions with phospholipid polar heads and the absence of glycoside/cholesterol interactions.

Moreover, MD simulations of cucumarioside A_7_ (**45**) interactions showed RMSD values with neighboring lipids was comparable to those observed during MD simulations in the control membrane system and for both did not exceed 2.34 Å. This result indicated no significant changes in lipid packaging induced by cucumarioside A_7_ (**45**); this is in good accordance with hemolytic activity, SAR data (Table 1), as well as other MD simulations which indicated the involvement of the aglycones side chain in the hydrophobic interactions with phospholipid fatty acid tails and cholesterol. It is obviously that hydroxyl groups in the side chain of **45** imped such interactions.

## 3. Materials and Methods

### 3.1. Model System for Artificial Plasma Membrane Mimicking the Erythrocyte Membrane

An asymmetric model bilayer comprising POPC (1-palmitoyl-2-oleoyl-sn-glycero-3-phosphocholine), CHOL (cholesterol), PSM (palmitoylshingomyeline for outer leaflet), or POPE (1-palmitoyl-2-oleoyl-sn-glycero-3-phosphoethanolamine) for the inner leaflet, respectively, in the ratio 1:2:1 was constructed by remote web resource CHARMM-GUIHMMM Builder [52,53], solvated, and equilibrated during 400 ns for optimal bilayer package.

### 3.2. Full Atom MD Simulations

Since we did not have any information on the possible orientation of glycosides during their interaction with the membrane, glycoside molecules were added to the previously equilibrated model membrane system and placed at a distance of 11 Å above the outer membrane leaflet. The orientation of the molecules was chosen arbitrarily provided that their long axis was located along the membrane surface (Appendix A). The model membrane simulation system with the glycosides was resolvated with water (25 Å above and below the membrane) and neutralized with counterions for a simulating box of 200 × 200 × 90 Å, protonated at pH 7.4, and the total potential energy of the systems was minimized with the energy gradient of 0.01 kcal/mol/Å^−1^ to remove initial unfavorable contacts, then heated from 0 to 300 K for 100 ns and equilibrated at 300 K for another 200 ns. 

The MD simulations of the free model membrane system or under the impact of glycosides in water environment were conducted with an Amber 14EHT force field. This was carried out with a checkpoint at 500 ps, a sample time of 10 ps, with Nosé-Poincaré–Andersen Hamiltonian equations of motion (NPA), and a time step of 0.001 ps, at a constant pressure (1 atm) and temperature (300 K) giving a total simulation time of 600 ns using MOE 2020.0901 CCG software [45]. Solvent molecules were treated as rigid. Computer simulations and theoretical studies were performed using cluster CCU *“Far Eastern computing resource”* FEB RAS (Vladivostok).

MD simulations of the control membrane system demonstrated RMSD value no higher than 2.34 Å.

The analysis of intramolecular interactions as well as the estimation of the interaction energy contribution was made with a ligand interaction suite from MOE 2020.0901 CCG software [45].

### 3.3. Triterpene Glycosides Chosen for MD Simulations

*Cucumarioside A_1_* (**40**): 3*β*-*O*-{3-*O-*methyl-*β*-d-xylopyranosyl-(1➝3)-*β*-d-glucopyranosyl-(1➝4)-*β*-d-quinovopyranosyl-(1➝2)-*β*-d-xylopyranosyl}-16*β*-acetoxyholosta-7,24-diene; mp 190 °C; [α]${}_{D}^{20}$–15° (*c* 0.1, C_5_H_5_N).

*Cucumarioside A_2_* (**59**): 3*β*-*O*-{3-*O*-methyl-*β*-d-xylopyranosyl-(1➝3)-*β*-d-glucopyranosyl-(1➝4)-*β*-d-quinovopyranosyl-(1➝2)-*β*-d-xylopyranosyl}-16*β*,24ξ-diacetoxyholosta-7,25-diene; mp 167°C; [α]${}_{D}^{20}$ –17 (*c* 0.1, C_5_H_5_N). HR ESI MS (+) *m/z*: 1179.5555 (calc 1179.5558) [M + Na]^+^.

*Cucumarioside A_7_* (**45**) is 3*β*-*O*-{3-*O*-methyl-*β*-d-xylopyranosyl-(1➝3)-*β*-d-glucopyranosyl-(1➝4)-*β*-d-quinovopyranosyl-(1➝2)-*β*-d-xylopyranosyl}-16*β*-acetoxyholosta-24*S*-hydroxy-7,25-diene; mp 183–185 °C; [α]${}_{D}^{20}$ –5 (*c* 0.1, C_5_H_5_N). HR ESI MS (+) *m/z*: 1137.5460 (calc 1137.5452) [M + Na]^+^.

*Cucumarioside A_8_* (**44**) 3*β*-O-[3-O-methyl-*β*-d-xylopyranosyl-(1➝3)-*β*-d-glucopyranosyl-(1➝4)-*β*-d-quinovopyranosyl-(1➝2)-*β*-d-xylopyranosyl]-16*β*-acetoxy-9*β*-H-lanosta-7,24-diene-18,20β-diol. mp 238–240 °C, [α]${}_{D}^{20}$–3 (c 0.1, C_5_H_5_N), HR MALDI TOF MS (+) *m/z*: 1125.5812 (calc 1125.5816) [M + Na]^+^.

## 4. Conclusions

The SAR for the sea cucumber triterpene glycosides illustrated by their action on mouse erythrocytes, is very complicated. Nevertheless, in our study, several clear trends were found, providing significant membranolytic activity for the glycosides, namely: the presence of a developed carbohydrate chain composed of four to six monosaccharide residues (with linear tetrasaccharide fragment) or a disaccharide chain with a sulfate group; the availability of 18(20)- or 18(16)-lactone and a normal (non-shortened) side chain; the presence of 9β-H, 7(8)-ene fragment or 9(11)-double bond. It was also observed that the influence of sulfate groups on the membranotropic action of the glycosides depends on the architecture of the sugar chain and the positions of sulfate groups. Hydroxyl groups attached to different positions of aglycone side chains extremely decrease the activity.

Using an in silico approach of full-atom MD simulations for the investigation of interactions of sea cucumber triterpene glycosides with the molecules composing the model lipid bilayer membrane has resulted in the clarification of several characteristics of the molecular mechanisms of membranolytic action of these compounds. It was revealed that the studied glycosides bound to the membrane surface mainly by hydrophobic interactions and hydrogen bonds, but the mode of such interactions depended on the aglycone side chain structure and varied to a great extent. The formation of multimolecular lipid/glycoside complexes led to membrane curvature followed by the subsequent membranolytic effects of the glycosides. Different mechanisms of glycoside/membrane interactions were discovered for cucumariosides A_1_ (**40**), A_8_ (**44**), and A_2_ (**59**). The first mechanism, inherent for **40** and **44,** was realized through the pore’s formation differed by the shape, stoichiometry, and the impact of diverse noncovalent interactions into complex assembling, depending on the glycoside structural peculiarities. The second mode of membranotropic action was realized by **59** through the formation of phospholipid and cholesterol clusters in the outer and inner membrane leaflets, correspondingly.

The observed peculiarities of membranotropic action are in good agreement with the corresponding data of in vitro hemolytic activity of the investigated compounds [28,29]. In fact, the hemolytic activity of pore-forming cucumariosides A_1_ (**40**) and A_8_ (**44**) were 0.07 and 0.70 µM/mL, correspondingly. The value for cluster-forming cucumarioside A_2_ (**59**) was 4.70 µM/mL, and cucumarioside A_7_ (**45**) demonstrating the weakest capacity to embed the membrane, was not active to the maximal studied concentration of 100.0 µM/mL.

Further in silico studies of the relationships of the membrane lipid composition and structural peculiarities of the glycosides demonstrating membranolytic activity are necessary to ascertain the molecular targets of glycoside/membrane bonding and to deepen the understanding of these complex multistage mechanisms.

## Figures and Tables

**Figure 1 marinedrugs-19-00604-f001:**
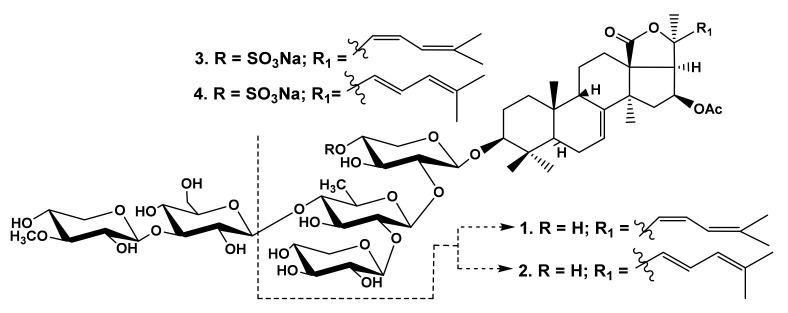
Structures of the glycosides **1**–**4** from *Eupentacta fraudatrix*.

**Figure 2 marinedrugs-19-00604-f002:**
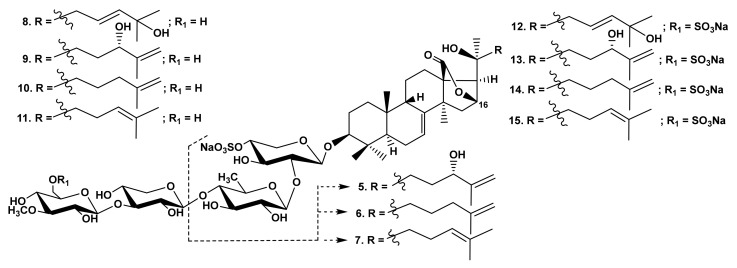
Structures of the glycosides **5**–**15** from *Massinum magnum*.

**Figure 3 marinedrugs-19-00604-f003:**
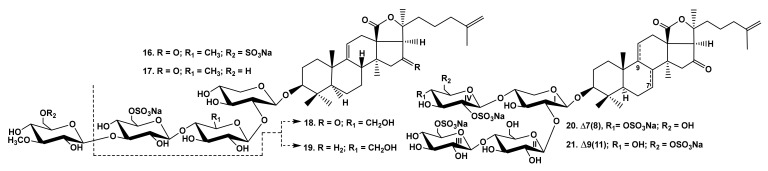
Structures of the glycosides **16**–**21** from *Psolus fabricii*.

**Figure 4 marinedrugs-19-00604-f004:**
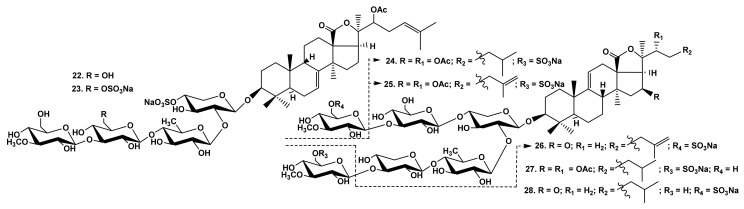
Structures of glycosides **22** and **23** from *Actinocucumis typica* and **24**–**28** from *Cladolabes shcmeltzii*.

**Figure 5 marinedrugs-19-00604-f005:**
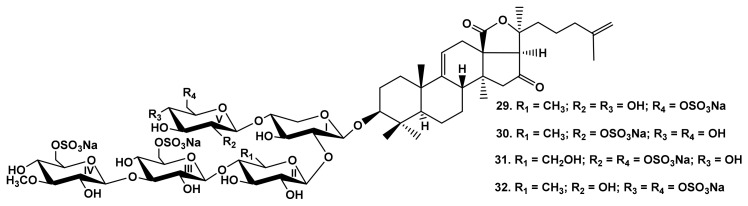
Structures of the glycosides **29**–**32** from *Psolus fabricii*.

**Figure 6 marinedrugs-19-00604-f006:**
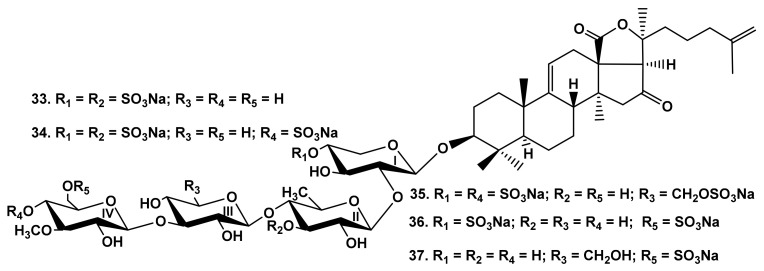
Structures of the glycosides **33**–**35** from *Colochirus quadrangularis*, **36** from *Colochirus robustus* and **37** from *Psolus fabricii*.

**Figure 7 marinedrugs-19-00604-f007:**
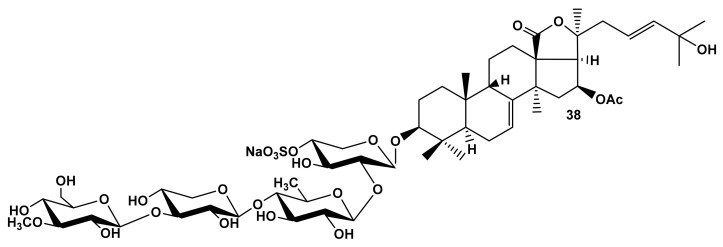
Structure of colochiroside B_2_ (**38**) from *Colochirus robustus*.

**Figure 8 marinedrugs-19-00604-f008:**
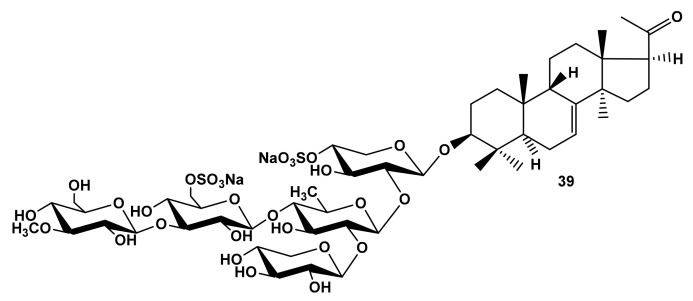
Structure of cucumarioside A_3_-2 from *Cucumaria fallax*.

**Figure 9 marinedrugs-19-00604-f009:**
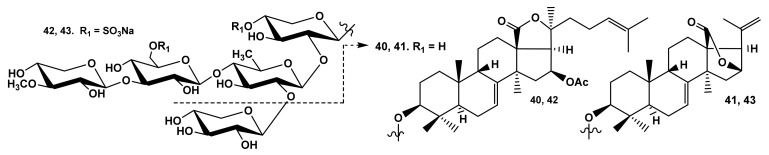
Structures of the glycosides **40**–**43** from *Eupentacta fraudatrix*.

**Figure 10 marinedrugs-19-00604-f010:**
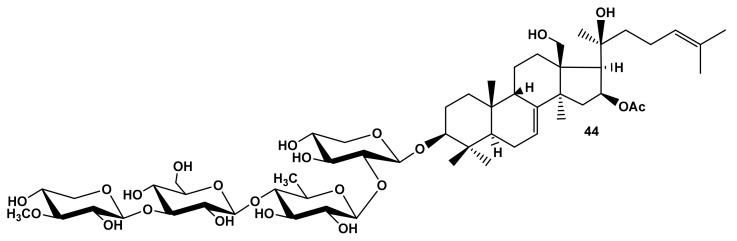
Structure of cucumarioside A_8_ (**44**) from *Eupentacta fraudatrix*.

**Figure 11 marinedrugs-19-00604-f011:**
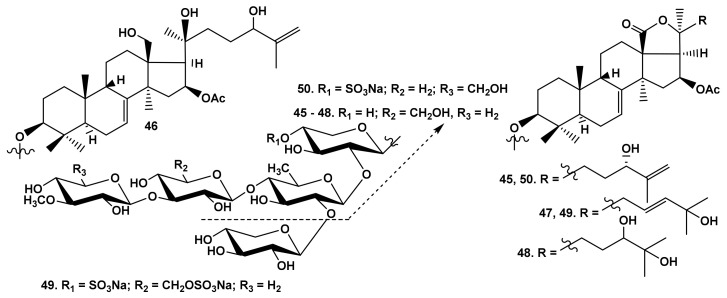
Structures of the glycosides **45**–**49** from *Eupentacta fraudatrix* and **50** from *Colochirus robustus*.

**Figure 12 marinedrugs-19-00604-f012:**
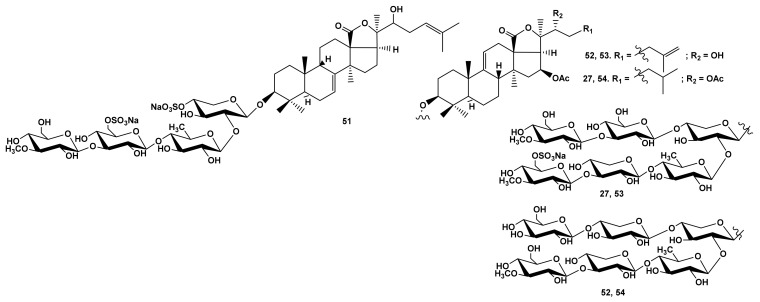
Structures of the glycosides **51**–**54** from *Cladolabes schmeltzii*.

**Figure 13 marinedrugs-19-00604-f013:**
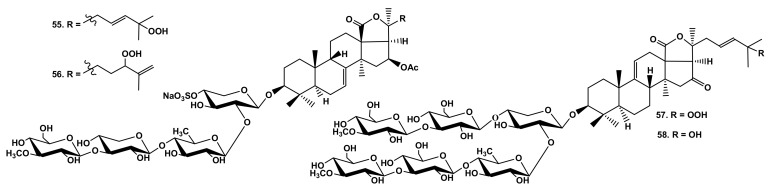
Structures of glycosides **55** and **56** from *Colochirus quadrangularis* and **57** and **58** from *Psolus fabricii*.

**Figure 14 marinedrugs-19-00604-f014:**
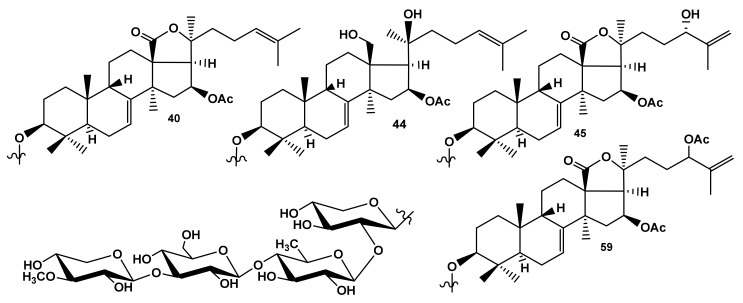
Structure of cucumariosides A_1_ (**40**), A_8_ (**44**), A_7_ (**45**), and A_2_ (**59**) used for in silico analysis of the interaction of the glycosides from the sea cucumber, *Eupentacta fraudatrix,* with the model membrane.

**Figure 15 marinedrugs-19-00604-f015:**
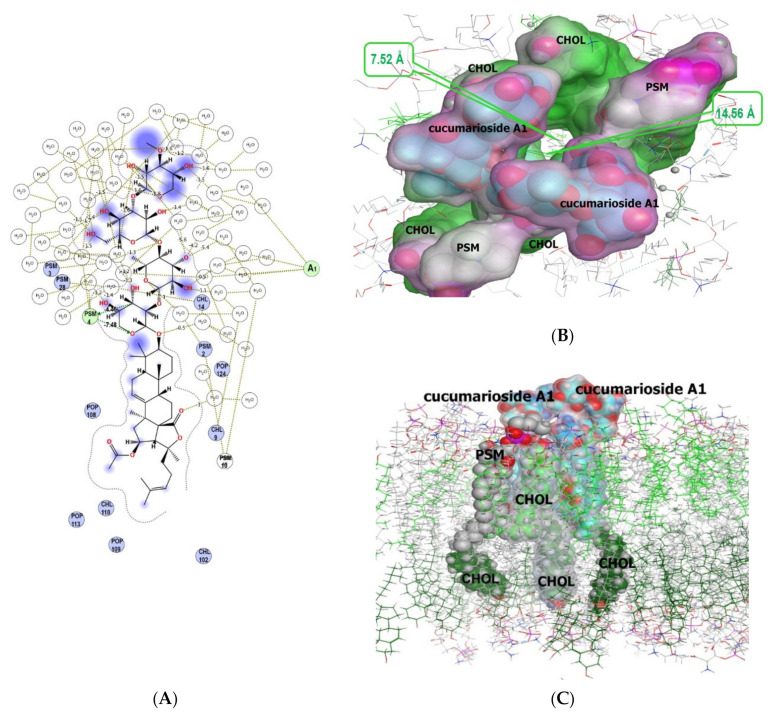
Spatial organization of multimolecular complex formed by two cucumarioside A_1_ (**40**) molecules (I and II) and the model membrane components. (**A**) 2D diagram of noncovalent intermolecular interactions of the glycoside with water-lipid environment. (**B**) Multimolecular complex is presented as a semitransparent molecular surface, colored according to its lipophilicity: hydrophilic areas are pink, lipophilic areas are green, the view is perpendicular to membrane surface. The molecules of solvent and some membrane components are deleted for simplicity. (**C**) Multimolecular complex in membrane environment, the view parallel to membrane surface. The glycoside is presented as cyan “ball” model, POPC+PSM and CHOL molecules (6 Å surrounding glycoside-lipid complex) of outer membrane leaflet are grey and light-green “ball” models, respectively; POPC+PSM and CHOL of inner membrane leaflet, distant from molecular assembly, are presented as grey and dark-green “ball and stick” models, respectively.

**Figure 16 marinedrugs-19-00604-f016:**
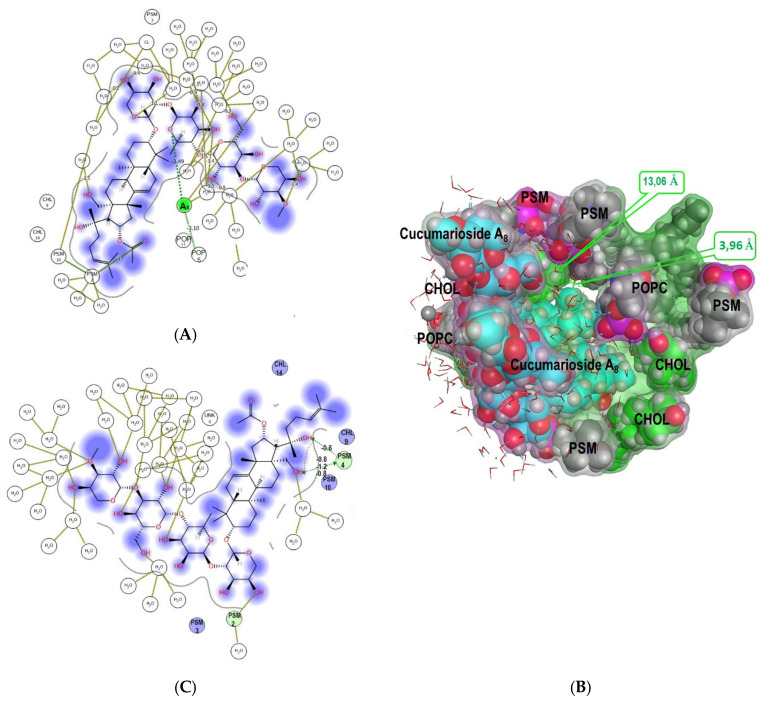
Spatial organization of multimolecular complex formed by two cucumarioside A_8_ (**44**) molecules (I and II) and the model membrane components. (**A**) 2D diagram of noncovalent intermolecular interactions of the glycoside with water-lipid environment. (**B**) Multimolecular complex is presented as a semitransparent molecular surface, colored according to its lipophilicity: hydrophilic areas are pink, lipophilic areas are green, the view is perpendicular to membrane surface. The glycoside is presented as cyan “ball” model, POPC+PSM and CHOL molecules (6 Å surrounding glycoside-lipid complex) of the outer membrane leaflet are grey and light-green “ball” models. The molecules of solvent and some membrane components are deleted for simplicity. (**C**) 2D diagram of noncovalent intermolecular interactions of cucumarioside A_8_ (**44**) with water-lipid environment at the initial stage of glycoside interaction with the model membrane.

**Figure 17 marinedrugs-19-00604-f017:**
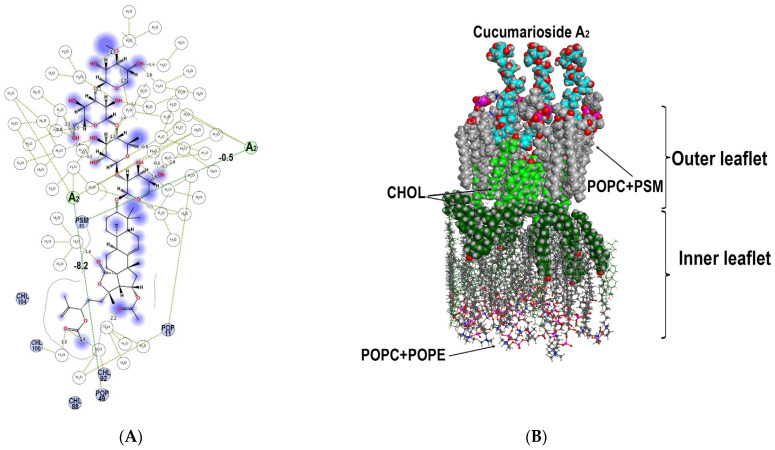
Spatial organization of multimolecular complex formed by three molecules (I–III) of cucumarioside A_2_ (**59**) and the components of model membrane. (**A**) 2D diagram of intermolecular noncovalent interactions of three cucumarioside A_2_ (**59**) molecules and the components of model water/lipid bilayer environment. Hydrogen bonds are green dotted lines. (**B**) Front view to the cucumarioside A_2_ (**59**) multimolecular complex with a model membrane. The glycoside is presented as cyan “ball” model, POPC+PSM and CHOL (6 Å surrounding glycoside) of the outer membrane leaflet are presented as grey and light-green “ball” models, respectively; POPC+POPE and CHOL of inner membrane leaflet, distant from multimolecular assembly, are presented as grey and dark-green “ball and stick” models, respectively; CHOL molecules of the inner membrane leaflet at 5 Å distant from multimolecular complex are presented as a dark-green “ball” model. The molecules of solvent and some membrane components are deleted for simplicity.

**Table 1 marinedrugs-19-00604-t001:** The hemolytic activities (synoptic data from the corresponding publications) of the glycosides **1**–**59** against mouse erythrocytes.

Glycoside	ED_50_, µM/mL	Glycoside	ED_50_, µM/mL	Glycoside	ED_50_, µM/mL
Cucumarioside B_1_ (**1**)	>100	Psolusoside K (**21**)	>100	Cucumarioside A_10_ (**41**)	20.00
Cucumarioside B_2_ (**2**)	18.8	Typicoside B_1_ (**22**)	0.33	Cucumarioside I_1_ (**42**)	23.24
Cucumarioside H_5_ (**3**)	3.2	Typicoside C_2_ (**23**)	0.18	Cucumarioside I_4_ (**43**)	75.00
Cucumarioside H (**4**)	3.8	Cladoloside I_1_ (**24**)	1.10	Cucumarioside A_8_ (**44**)	0.70
Magnumoside A_2_ (**5**)	33.33	Cladoloside I_2_ (**25**)	2.04	Cucumarioside A_7_ (**45**)	>100
Magnumoside A_3_ (**6**)	12.53	Cladoloside J_1_ (**26**)	1.37	Cucumarioside A_9_ (**46**)	>100
Magnumoside A_4_ (**7**)	20.12	Cladoloside K_1_ (**27**)	0.18	Cucumarioside A_11_ (**47**)	>100
Magnumoside B_1_ (**8**)	49.57	Cladoloside L_1_ (**28**)	0.82	Cucumarioside A_14_ (**48**)	>100
Magnumoside B_2_ (**9**)	58.11	Psolusoside L (**29**)	2.42	Cucumarioside I_3_ (**49**)	>100
Magnumoside B_3_ (**10**)	8.49	Psolusoside M (**30**)	67.83	Colochiroside B_1_ (**50**)	39.5
Magnumoside B_4_ (**11**)	1.42	Psolusoside Q (**31**)	>100	Typicoside C_1_ (**51**)	6.25
Magnumoside C_1_ (**12**)	6.97	Psolusoside P (**32**)	10.92	Cladoloside D_2_ (**52**)	10.40
Magnumoside C_2_ (**13**)	16.20	Quadrangularisoside B_2_ (**33**)	0.51	Cladoloside K_2_ (**53**)	11.41
Magnumoside C_3_ (**14**)	17.80	QuadrangularisosideD_2_ (**34**)	3.31	Cladoloside D_1_ (**54**)	0.67
Magnumoside C_4_ (**15**)	6.52	QuadrangularisosideE (**35**)	2.04	QuadrangularisosideA (**55**)	1.57
Psolusoside A (**16**)	1.4	Colochiroside C (**36**)	2.5	QuadrangularisosideA_1_ (**56**)	1.11
Psolusoside E (**17**)	0.23	Psolusoside F (**37**)	2.8	Psolusoside D_3_ (**57**)	1.12
Psolusoside H (**18**)	2.5	Colochiroside B_2_ (**38**)	37.02	Psolusoside D_5_ (**58**)	12.37
Psolusoside H_1_ (**19**)	2.7	Cucumarioside A_3_-2 (**39**)	40.6	Cucumarioside A_2_ (**59**)	4.70
Psolusoside J (**20**)	>100	Cucumarioside A_1_ (**40**)	0.07		

**Table 2 marinedrugs-19-00604-t002:** Noncovalent intermolecular interactions inside the multimolecular complex formed by two molecules (I and II) of cucumarioside A_1_ (**40**) and components of model lipid bilayer membrane.

Type of Bonding	CucumariosideA_1_ (40) Molecule	Membrane Component	Energy Contribution, kcal/mol	Distance, Å
Hydrogen bond	I	PSM4	−11.94	4.05
Hydrophobic	I	PSM4	−0.5	3.31
Hydrophobic	I	POPC108	−7.21	3.93
Hydrophobic	I	PSM2	−5.52	4.13
Hydrophobic	I	POP109	−4.69	3.92
Hydrophobic	I	PSM10	−3.71	4.19
Hydrophobic	I	CHOL9	−3.69	4.13
Hydrophobic	I	CHOL14	−2.18	4.01
Hydrophobic	I	POPC124	−1.59	4.02
Hydrophobic	I	POPC113	−0.55	4.13
Hydrophobic	II	CHOL38	−11.05	4.07
Hydrophobic	II	PSM31	−10.82	4.08
Hydrophobic	II	POPC124	−8.38	4.11
Hydrophobic	II	CHOL46	−4.77	4.06
Hydrophobic	II	CHOL14	−4.50	3.93
Hydrophobic	II	PSM28	−1.06	4.15
Hydrophobic	II	PSM74	0.05	3.95

**Table 3 marinedrugs-19-00604-t003:** Noncovalent intermolecular interactions inside multimolecular complex formed by two molecules (I, II) of cucumarioside A_8_ (**44**) and the components of model lipid bilayer membrane.

Type of Bonding	Cucumarioside A_8_ (44) Molecule	Membrane Component	Energy Contribution, kcal/mol	Distance, Å
Hydrogen bond	II	I	−3.49	3.36
Hydrophobic	II	I	−8.75	3.95
Hydrophobic	II	PSM20	−12.41	4.03
Hydrophobic	I	PSM2	−8.60	4.07
Hydrophobic	II	POPC13	−7.93	3.97
Hydrophobic	II	CHL7	−7.20	4.02
Hydrophobic	II	PSM2	−4.28	4.04
Hydrophobic	I	CHL9	−4.06	4.06
Hydrophobic	I	PSM10	−3.91	4.08
Hydrophobic	II	POPC108 *	−3.72	3.94
Hydrophobic	II	CHL14	−3.23	4.11
Hydrogen bond	II	POPC5	−3.10	2.60
Hydrophobic	I	PSM3	−2.31	3.96
Hydrophobic	II	POPC113 *	−2.02	4.21
Hydrophobic	I	POPC13	−1.39	3.59
Hydrophobic	II	PSM28	−1.01	4.26
Hydrogen bond	I	PSM2	−1.00	3.01

*—the inner membrane leaflet.

**Table 4 marinedrugs-19-00604-t004:** Noncovalent intermolecular interactions inside multimolecular complex formed by three molecules (I–III) of cucumarioside A_2_ (**59**) and components of model lipid bilayer membrane.

Type of Bonding	Cucumarioside A_2_ (59) Molecule	Membrane Component	Energy Contribution, kcal/mol	Distance, Å
Hydrophobic	I	PSM51	−4.63	4.21
Hydrophobic	I	POPC11	−3.34	3.99
Hydrophobic	I	CHOL92	−0.63	3.89
Hydrophobic	I	POPC49	−1.23	3.99
Hydrogen bond	II	PSM51	−0.49	3.18
Hydrophobic	II	PSM57	−6.19	4.14
Hydrophobic	II	CHOL104	−6.1	3.98
Hydrophobic	II	PSM55	−3.3	4.07
Hydrophobic	II	POPC11	−2.78	4.17
Hydrophobic	II	PSM51	−2.18	4.08
Hydrogen bond	III	POPC49	−8.2	2.49
Hydrophobic	III	POPC11	−3.08	4.20
Hydrophobic	III	POPC49	−1.43	3.91
Hydrophobic	III	CHOL99	−0.67	3.53

## Data Availability

Not applicable.

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
