# Peer review of "Structure-Activity Relationships of Holothuroid’s Triterpene Glycosides and Some In Silico Insights Obtained by Molecular Dynamics Study on the Mechanisms of Their Membranolytic Action"

_marinedrugs, 2021, doi:10.3390/md19110604_

Round 1

Reviewer 1 Report

The article ''Structure-Activity Relationships of Holothuroid’s Triterpene Glycosides and Some in silico Insights on the Mechanisms of their Membranolytic Action'' is a nice piece of work 

I have here some comments that may need a respond:

1- The title should have ''molecular dynamics study''

2- Line 13-17 in the abstract need to move to discussion part. The abstract should be concise and concentrate on findings.

3- Line 42 in introduction: ''Some trends of SAR...have been discussed [5.6]'' , what are these trends and are they related to the present compounds?

4- Aim of the study should be clarified; is it SAR, or study of molecular interactions and MD or both

5- The unit of energy in table 2 is ''Kcal/mol'' not Kcal/M''

6- The RMSD vs time graph should be generated to show the equilibrium of MD and at what time interval was achieved

7- The method of MD should be described in details and how the author analyzed the trajectory file

8- The author mentioned that compounds were introduced randomly in MD simulations, how ?

9- By using MOE the author performed a MD study at 100ns, 200ns and 600 ns how did you observe and analyze the conformational output and did you make conformational clustering?

Author Response

The article ''Structure-Activity Relationships of Holothuroid’s Triterpene Glycosides and Some in silico Insights on the Mechanisms of their Membranolytic Action'' is a nice piece of work

I have here some comments that may need a respond:

1- The title should have ''molecular dynamics study''

Reply:

The title was changed according to the Referee 1 comment: ''Structure-Activity Relationships of Holothuroid’s Triterpene Glycosides and Some in silico Insights Obtained by Molecular Dynamics Study on the Mechanisms of their Membranolytic Action'

2- Line 13-17 in the abstract need to move to discussion part. The abstract should be concise and concentrate on findings.

Reply:

Lines 13-17 The phrase “The structure/activity dependence has revealed complicated and ambiguous character because, from the one hand, the impact in the membranotropic action of a certain structural element depends on the combination of such elements in the glycoside molecule and, from the other hand, the activity depends on the membrane lipid composition” was removed from the abstract.

3- Line 42 in introduction: ''Some trends of SAR...have been discussed [5.6]'', what are these trends and are they related to the present compounds?

Reply:

In the references 5, 6 the in vitro and in vivo biological activity of the glycosides from different sea cucumber species are reviewed and summarized showing the influence of structural peculiarities of their aglycones and carbohydrate chains to the effect demonstrated by them, also making evident diverse mechanisms of their action due to they can affect the set of molecular targets. So, these papers discuss the SAR of the sea cucumbers triterpene glycosides as the class of compounds possessing great structural variability while maintaining the general plan of structure. These references are relevant in the context of our manuscript showing the gaps in the understanding of the SAR and mechanisms of their physiologic action despite the extensive studies of these compounds.

4- Aim of the study should be clarified; is it SAR, or study of molecular interactions and MD or both

Reply:

The aims of our research were specified: “The aims of this study were: the analysis of SAR data for broad series of the sea cucumber glycosides obtained mainly by our research team during last years on different tumor cell lines and erythrocytes and also the explanation of these data by modelling the interactions of the glycosides from the sea cucumber Eupentacta fraudatrix with the constituents of model red blood cell membrane with the full-atom molecular dynamics (MD) simulation.”

5- The unit of energy in table 2 is ''Kcal/mol'' not Kcal/M''

Reply:

It is fixed.

6- The RMSD vs time graph should be generated to show the equilibrium of MD and at what time interval was achieved

Reply:

The aim of this study was to search for a possible relationship between the structure of the glycoside and the character/manner of its interaction with the model membrane. For this purpose, from the list of compounds presented in Table 1, the 4 glycosides were selected, differing in structure and hemolytic activity significantly, MD simulations were carried out. This is the first stage. In the future, we plan to form groups of glycosides with small variations in their structure and conduct a more detailed study of the process of their incorporation into the membrane. And in this case (at the next stage of the study), a conformational clustering and the time of the equilibrium achievement will be of fundamental importance, since they will allow a justified/reasonable correlation between the mechanism (with relaxation time as one of parameters) of incorporation into the membrane, the structural features of the glycoside molecule and its biological activity. We thank the reviewer for valuable advice.

7- The method of MD should be described in details and how the author analyzed the trajectory file

Reply:

The section 3.2. extended with MD parameters: The MD simulations of the free model membrane system or under the impact of glycosides in water environment were conducted in Amber 14EHT force field, with a checkpoint at 500 ps, sample time of 10 ps, with Nosé-Poincaré-Andersen Hamiltonian equations of motion (NPA), and a time step of 0.001 ps, at the constant pressure (1 atm) and temperature (300 K) with giving a total simulation time of 600 ns using МОЕ 2020.0901 CCG software [43]. Solvent molecules were treated as rigid. The trajectory of Potential energy U (r) of the system at time t was exploited to monitor the MD course.

8- The author mentioned that compounds were introduced randomly in MD simulations, how?

Reply:

We agree that the phrase in section 3.2."The glycoside molecules were randomly added to the equilibrated model membrane system and placed at 11 Å distance from the membrane surface." requires clarification. Indeed, by using the word “randomly” we meant an arbitrary choice of the glycoside molecule orientation provided that its long axis is located along the membrane surface at a distance of 11 Å. This phrase was replaced by: “Since we did not have any information about the possible orientation of glycosides during their interaction with the membrane, glycoside molecules were added to the previously equilibrated model membrane system and placed at a distance of 11 Å above the outer membrane leaflet, while the orientation of the molecules was chosen arbitrarily provided that their long axis is located along the membrane surface. For example, the initial conformation of cucumarioside A8 (44) is presented in figure S2 A).” The corresponding Figure S2 was added to the Supplementary materials.

9- By using MOE the author performed a MD study at 100ns, 200ns and 600 ns how did you observe and analyze the conformational output and did you make conformational clustering?

Reply:

MOE software allows one to observe and to analyze the conformation during MD simulations with a graphical interface, and also generates the output Potential energy U (r) of the system at time t to monitor the system state. We carried out preliminary preparation of the structures of the glycosides studied, which included structure Washing, Protonation at pH 7.4, Energy Minimization, and Stochastic Conformational Search with appropriate suits in МОЕ 2020.0901 CCG. The section «2.2.5. Correlation analysis» has been expanded: “2.2.5. Correlation analysis. To determine the structural elements of glycosides, which might be responsible for membrane recognition а set of physical properties of fifty-nine glycosides (represented in the Table 1) were analyzed with МОЕ 2020.0901 CCG software [45]. Models of the spatial structure of the studied glycosides were built, protonated at pH 7,4 and subjected to energy minimization and conformational search with corresponding/appropriate МОЕ suits, and the dominant glycoside conformations were selected. The numerical descriptions or characterizations of the molecules, that provide their physical properties, such as the octanol/water partition coefficient, the polar surface area, the van der Waals (VDW) volume, approximation to the sum of VDW surface areas of pure hydrogen bond acceptors/donors, the approximation to the sum of VDW surface areas of hydrophobic/polar atoms etc. (in total 296), as well as their correlation matrix were calculated with the QuaSAR-Descriptor tool of МОЕ 2020.0901 CCG software [45] (Figure S1)”.

Reviewer 2 Report

This manuscript is well-conceived and very interesting.

Abstract is too long, please summarize and simplify the following paragraph “ Two different mechanisms of glycoside/membrane interactions were discovered: one  of them realized through the pore formation (by cucumariosides A1 (40) and A8 (44)) that is preceded by bonding of the glycosides with membrane sphingomyelin. Further incorporation of the glyco-  sides into the leaflet is provided by hydrophobic interactions of the aglycones with phospholipids, sphingomyelin and cholesterol. However, the stoichiometry of the multimolecular complexes  formed by cucumariosides A1 (40) and A8 (44) as well as non-covalent intermolecular interactions inside complexes were different. The second mechanism realized by cucumarioside A2 (59) through  the formation of phospholipid and cholesterol clusters in the outer and inner membrane leaflets, correspondingly”

The sentence “In silico technique was applied to reinforce the numerous experimental observations (SAR) by the modelling a multitude of inter-molecular interactions at a high spatial (atomic level) and  temporal (nanosecond) resolution within a simulation framework that can reconstitute the natural behavior on the basis of physical interactions” should be accompanied by more references such as follows:

Molecular Diversity Volume 20, Issue 1, Pages 77 - 921 February 2016

Molecules Open AccessVolume 23, Issue 1 2018 Article number 120

Please, improve quality of figure 15 (B) and of fig 17 (B)

Distances shown in tables 2-3-4 should be reported as “.” Å

Please, better explain the sentence “The glycoside molecules were randomly added to the equilibrated model membrane  system and placed at 11 Å distance from the membrane surface.”

Author Response

Referee report 2.
This manuscript is well-conceived and very interesting.

1- Abstract is too long, please summarize and simplify the following paragraph “ Two different mechanisms of glycoside/membrane interactions were discovered: one of them realized through the pore formation (by cucumariosides A1 (40) and A8 (44)) that is preceded by bonding of the glycosides with membrane sphingomyelin. Further incorporation of the glycosides into the leaflet is provided by hydrophobic interactions of the aglycones with phospholipids, sphingomyelin and cholesterol. However, the stoichiometry of the multimolecular complexes formed by cucumariosides A1 (40) and A8 (44) as well as non-covalent intermolecular interactions inside complexes were different. The second mechanism realized by cucumarioside A2 (59) through the formation of phospholipid and cholesterol clusters in the outer and inner membrane leaflets, correspondingly”

Reply:

The following paragraphs in the abstract were shortened: “Two different mechanisms of glycoside/membrane interactions were discovered. The first one realized through the pore formation (by cucumariosides A1 (40) and A8 (44)), preceded by bonding of the glycosides with membrane sphingomyelin, phospholipids and cholesterol. Non-covalent intermolecular interactions inside multimolecular membrane complexes and their stoichiometry differed for 40 and 44.”

2- The sentence “In silico technique was applied to reinforce the numerous experimental observations (SAR) by the modelling a multitude of inter-molecular interactions at a high spatial (atomic level) and temporal (nanosecond) resolution within a simulation framework that can reconstitute the natural behavior on the basis of physical interactions” should be accompanied by more references such as follows:

Molecular Diversity Volume 20, Issue 1, Pages 77 - 921 February 2016

Molecules Open AccessVolume 23, Issue 1 2018 Article number 120

Reply:

The references [20, 21] were added to sentence “In silico technique was applied to reinforce the numerous experimental observations (SAR) by the modelling a multitude of inter-molecular interactions at a high spatial (atomic level) and temporal (nanosecond) resolution within a simulation framework that can reconstitute the natural behavior on the basis of physical interactions [20, 21]”. [20] Guariento, S.; Bruno, O.; Fossa, P.; Cichero, E. New insights into PDE4B inhibitor selectivity: CoMFA analyses and molecular docking studies. Mol. Divers. 2016, 20, 77–92. https://doi.org/10.1007/s11030-015-9631-1; [21] Rusnati, M.; Sala, D.; Orro, A.; Bugatti, A.; Trombetti, G.; Cichero, E.; Urbinati, C.; Di Somma, M.; Millo, E.; Galietta, L.J.V.; Milanesi, L.; Fossa, P.; D’Ursi, P. Speeding Up the Identification of Cystic Fibrosis Transmembrane Conductance Regulator-Targeted Drugs: An Approach Based on Bioinformatics Strategies and Surface Plasmon Resonance. Molecules 2018, 23, 120. https://doi.org/10.3390/molecules23010120. The references numbering was changed in accordance to the insertion of additional [20, 21].

3- Please, improve quality of figure 15 (B) and of fig 17 (B)

Reply:

The figures 15 (B) and of 17 (B) have been improved.

4- Distances shown in tables 2-3-4 should be reported as “.” Å

Reply:

Corresponding changes were made to tables 2-3-4

5- Please, better explain the sentence “The glycoside molecules were randomly added to the equilibrated model membrane system and placed at 11 Å distance from the membrane surface.”

Reply:

We agree that the phrase in section 3.2."The glycoside molecules were randomly added to the equilibrated model membrane system and placed at 11 Å distance from the membrane surface." requires clarification. Indeed, by using the word “randomly” we meant an arbitrary choice of the glycoside molecule orientation provided that its long axis is located along the membrane surface at a distance of 11 Å. This phrase was replaced by: “Since we did not have any information about the possible orientation of glycosides during their interaction with the membrane, glycoside molecules were added to the previously equilibrated model membrane system and placed at a distance of 11 Å above the outer membrane leaflet, while the orientation of the molecules was chosen arbitrarily provided that their long axis is located along the membrane surface. For example, the initial conformation of cucumarioside A8 (44) is presented in figure S2 A)”. The corresponding Figure S2 was added to the Supplementary materials.

Reviewer 3 Report

In the manuscript (ID: marinedrugs-1417267), the authors of Zelepuga et al studied structure-activity relationships (SAR) for broad series of the sea cucumber glycosides on different tumor cell lines and erythrocytes and in silico modulation of interaction of selected glycosides from the sea cucumber Eupentacta fraudatrix with model erythrocyte membranes using full-atom molecular dynamics (MD) simulations.

The manuscript is interesting, however it should be improved.

I have a few minor concerns that can be easily addressed:

1) In the section 3 “Materials and methods” there is lack of information about glycosides that were chosen for studies. The authors describe glycosides firstly in the section “Results and Discussion”.

2) The authors did not explain the methodology on how the glycosides bind to the membrane, whether during dynamics they are subject to forces that bind them to the membrane or  were added earlier to the membrane.

3)  The section 2.2.5 the authors describe the correlation analysis. However, no data were shown, thus it’s difficult to rate such analysis. In my opinion the physical properties of glycosides should be included in supplementary materials.

Author Response

Referee report 3

In the manuscript (ID: marinedrugs-1417267), the authors of Zelepuga et al studied structure-activity relationships (SAR) for broad series of the sea cucumber glycosides on different tumor cell lines and erythrocytes and in silico modulation of interaction of selected glycosides from the sea cucumber Eupentacta fraudatrix with model erythrocyte membranes using full-atom molecular dynamics (MD) simulations.

The manuscript is interesting, however, it should be improved.

I have a few minor concerns that can be easily addressed:

1) In the section 3 “Materials and methods” there is lack of information about glycosides that were chosen for studies. The authors describe glycosides firstly in the section “Results and Discussion”.

Reply:

The corresponding information is added. The authors describe glycosides firstly in the section “Results and Discussion”. The systematic names of the glycosides chosen for in silico studies, their values of melting points and specific rotations as well as the data of high-resolution mass-spectrometry were added to the section 3.3 “Triterpene glycosides chosen for MD simulations” of “Materials and methods”.

2) The authors did not explain the methodology on how the glycosides bind to the membrane, whether during dynamics they are subject to forces that bind them to the membrane or were added earlier to the membrane.

Reply:

“Since we did not have any information about the possible orientation of glycosides during their interaction with the membrane, glycoside molecules were added to the previously equilibrated model membrane system and placed at a distance of 11 Å above the outer membrane leaflet, while the orientation of the molecules was chosen arbitrarily provided that their long axis is located along the membrane surface (Figure S2 A).” The section 2.3.2. was changed to: “The in silico study of the action of cucumarioside A8 (44) from E. fraudatrix [29] on a model erythrocyte membrane with MD simulations evidenced that the process apparently accrues in several stages: driven by electrostatic attracting the glycoside reaches the membrane with its carbohydrate part and can anchor to phospholipid polar heads through hydrogen bonds (Figure 16 C, S2), after that its aglycone moiety is completely immerses into the lipid layer as well as the multimolecular assembly rearranges. Moreover, our computational results have disclosed the feasibility of the glycoside to induce the “pore-like” complex formation inside the membrane having stoichiometry of glycoside/CHOL/POPC/PSM (2/3/2/5) (Figure 16 A, B, Table 3)”.

3) The section 2.2.5 the authors describe the correlation analysis. However, no data were shown, thus it’s difficult to rate such analysis. In my opinion the physical properties of glycosides should be included in supplementary materials.

Reply:

The Correlation matrix (Figure S1) was added to supplementary materials. Figure S1. The Correlation matrix of the hemolytic activities of glycosides in vitro (ED50, µM/mL, Table 1) and certain calculated molecular 2D and 3D descriptors conducted with the QuaSAR-Descriptor tool of МОЕ 2020.0901 CCG software [1]. moderate positive correlation of their activity with the atomic contribution to Log of the octanol/water partition coefficient (h_logP) [2], the total negative VDW surface area (Å2), the number of oxygen atoms (a_no), the atomic valence connectivity index (chi0v), kappa shape indexes (Kier) [3], describing different aspects of molecular shape, the molecular VDW volume (Vol, vdw_vol, VSA_acc, (Å3)) were disclosed.

  1. Molecular Operating Environment (MOE), 2020.09; Chemical Computing Group ULC, 1010 Sherbrooke St. West, Suite #910, Montreal, QC, Canada, H3A 2R7, 2020.
  2. Wildman, S.A.; Crippen, G.M. Prediction of Physiochemical Parameters by Atomic Contributions. Chem. Inf. Comput. Sci. 1999, 39, 868–873.
  3. Hall, L.H.; Kier, L.B. The Molecular Connectivity Chi Indices and Kappa Shape Indices in Structure-Property Modeling. Comput. Chem.1991, 2, 367−422. DOI: 10,1002/9780470125793,ch9